# Evaluation of Surgical Approaches and Use of Adjuvant Radiotherapy with Respect to Oncologic Outcomes in the Management of Clinically Early-Stage Cervical Carcinoma

Laura Burgess [1,2,*], Wafa AlDuwaisan [3], Tinghua Zhang [4], Krystine Lupe [1,2], Michael Fung-Kee-Fung [3], Wylam Faught [3], Tien Le [3] and Rajiv Samant [1,2]

1 Department of Radiology, Division of Radiation Oncology, University of Ottawa, Ottawa, ON K1H 8L6, Canada
2 Department of Radiology, Division of Radiation Oncology, The Ottawa Hospital Research Institute, Ottawa, ON K1H 8L6, Canada
3 Department of Obstetrics and Gynecology, Division of Gynecology Oncology, University of Ottawa, Ottawa, ON K1H 8L6, Canada
4 The Ottawa Methods Centre, The Ottawa Hospital Research Institute, Ottawa, ON K1H 8L6, Canada
* Correspondence: lburgess@toh.ca

**Abstract:** The standard of care for early-stage cervix cancer is radical hysterectomy with pelvic lymphadenectomy. Adjuvant radiotherapy (RT) or chemoradiotherapy may be administered to reduce the risk of recurrence in patients considered to be at elevated risk based on a combination of pathologic factors. We performed a retrospective review to determine oncologic outcomes in patients treated for early-stage cervix cancer and to determine if surgical approach impacted oncologic outcomes or the decision to use adjuvant therapy. In total, 174 women underwent radical hysterectomy and pelvic lymphadenectomy over the 15-year period. Most of these women (146) had open surgery and 28 had minimally invasive surgery (MIS). In total, 81 had adjuvant pelvic RT; 76 in the open surgery group (52%) and 5 in the MIS group (18%). Five-year PFS and OS, respectively, were 84% and 91%. Five-year PFS was significantly lower in patients who had MIS vs. open surgery, without a difference in 5-year OS, suggesting MIS should be avoided. Five-year PFS was the same with RT or with its omission, despite those treated with RT having higher risk disease. We have demonstrated excellent outcomes in patients with early-stage cervix cancer after primary surgery and selective use of RT, with few recurrences and excellent survival.

**Keywords:** cervix cancer; radiotherapy; minimally invasive surgery; robotic-assisted; adjuvant therapy; outcomes; recurrence risk

## 1. Introduction

Cervix cancer is the third most commonly diagnosed gynecologic cancer, with an estimated 1450 Canadian women diagnosed and 380 estimated deaths in 2022 [1]. Radical hysterectomy with pelvic lymphadenectomy is often recommended for patients with early-stage cervix cancer. The procedure may be performed via laparotomy (open), laparoscopy, or robotic-assisted surgical approaches. Over the past 10 years, there has been a gradual shift to a minimally invasive (MIS; laparoscopy or robotic-assisted) approach because of perceived lower operative morbidities and improved perioperative outcomes compared to an open approach [2–4]. However, in 2018 a prospective randomized trial and large population database review demonstrated that an MIS approach resulted in a significantly lower disease-free survival and overall survival than open surgery among women with early-stage cervix cancer [5,6].

Adjuvant radiotherapy (RT) or chemoradiotherapy (chemoRT) may be considered to reduce the risk of recurrence in patients considered to be at elevated risk based on a

combination of pathologic factors [7,8]. Sedlis et al. demonstrated that adjuvant pelvic radiotherapy reduces recurrences in women with stage 1B cervix cancer meeting at least two of the following criteria: primary greater than 4cm in size, stromal invasion > 1/3 and lymphovascular invasion (LVSI) [7]. After median follow-up 8.4 years, use of RT (given for two or more of criteria) was associated with significant reduction in risk of recurrence (HR 0.54; 90% CI 0.35–0.81) and risk of progression or death (HR 0.58; 90% CI 0.40–0.85) [9]. More recently, the four-factor model was introduced. This uses Sedlis criteria but also incorporates histology, with adenocarcinoma conferring additional recurrence risk, to determine whether adjuvant radiotherapy is indicated [10]. Specifically, it determined that the presence of any two of the four factors predicted for recurrence. Similarly, Peters et al. identified that the addition of cisplatin-based chemotherapy to pelvic radiotherapy in the presence of any of lymph node involvement, positive margins or positive parametria confers both a progression-free survival (17% benefit at 4 years with chemoRT vs. RT alone) and overall survival benefit (10% benefit at 4 years with chemoRT vs. RT alone) [8].

The aim of the study was to determine oncologic outcomes in surgically treated patients with early-stage cervix carcinoma at our institution; specifically, to determine oncologic outcomes, the impact of a surgical approach and the impact of use of adjuvant therapy. Additionally, we aimed to see if the use of adjuvant therapy was primarily driven by Sedlis and Peters criteria.

## 2. Materials and Methods

### 2.1. Design

Institutional research ethics board approval was obtained. A retrospective review of all patients with cervix cancers operated on by the division of gynecologic oncology at the Ottawa Hospital from 1 June 2003 to 31 July 2018. Patients treated for early-stage cervix cancer treated who underwent radical hysterectomy and pelvic lymphadenectomy with or without adjuvant therapy were identified. Most patients did not undergo routine pre-operative CT or MRI imaging of the abdomen or pelvis, as per local institutional practice. The choice of surgical approach, open or MIS (either robotic or laparoscopy) was the discretion of the surgeon. Exclusion criteria included patients treated with trachelectomy, neoadjuvant chemoRT, primary chemoRT or were found to have squamous intraepithelial lesions rather than invasive cancer.

Demographics, tumor related characteristics, surgical approach, adjuvant treatments (radiotherapy and chemotherapy) and detailed clinical outcomes including local, regional, distant recurrences and death were collected by retrospective chart reviews. Intra-operative, post-operative complications and readmission to hospital within thirty days were also abstracted.

### 2.2. Analysis

Descriptive statistics were used to summarize patient demographics and perioperative outcomes. Categorical variables were compared using Chi-Square test or Fisher's Exact test where appropriate. Continuous variables were compared across treatment groups with ANOVA tests. Fisher's Exact test was used to compare the difference in the proportion of peri/postoperative complications, and risk for emergency room visit/readmission within 30 days of the procedure between surgical approaches.

Time to Recurrence, progression-free survival (PFS), and overall survival (OS) were estimated with Kaplan–Meier method. The log-rank test was used to assess significance and 5-year PFS and 5-year OS are reported. All $p$-values < 0.05 were considered statistically significant.

## 3. Results

Over the 15-year period, 174 women underwent radical hysterectomy and pelvic lymphadenectomy. Patient demographics are summarized in Table 1. Their median age was 43 years (range 22–79). Most had squamous cell carcinoma (54%), with most of the remaining having adenocarcinoma (40.2%) and only 5.7% other pathology. As per FIGO

2009 staging, most patients had stage 1A or 1B1 disease (71.2%), but 11.5% had 1B2 and 17.2% had a higher stage on final pathology. Of these, four had stage II disease and the others were node positive on pathology.

**Table 1.** (**A**) Baseline characteristics in 174 patients with respect to surgical approach. Sedlis 4-factor model and Peters criteria are not exclusive and patients may have met more than one of these criteria. (**B**). Baseline characteristics in 174 patients with respect to use of adjuvant RT. Sedlis 4-factor model and Peters criteria are not exclusive and patients may have met more than one of these criteria. (**C**) Baseline characteristics in 146 patients treated with open surgery with respect to the use of adjuvant RT. Sedlis 4-factor model and Peters criteria are not exclusive and patients may have met more than one of these criteria. Statistically significant *p*-values are bolded.

| A | | | | |
|---|---|---|---|---|
| | **Total (174)** | **MIS (28)** | **Open (146)** | ***p*-Value** |
| Age (years, median, range) | 43 (22–79) | 41 (24–57) | 43 (22–79) | 0.13 |
| F/U (years, median, range) | 4.1 (0–14.8) | 3.4 (0–7.9) | 4.3 (0.1–14.8) | 0.17 |
| Histopathology | | | | **0.016** |
| Adenocarcinoma | 70 (40.2%) | 18 (64.3%) | 52 (35.6%) | |
| Squamous cell carcinoma | 94 (54.0%) | 10 (35.7%) | 84 (57.5%) | |
| Other | 10 (5.7%) | 0 (0%) | 10 (6.8%) | |
| Final stage | | | | **0.003** |
| 1A1/2 | 23 (13.2%) | 7 (25%) | 16 (11%) | |
| 1B1 | 100 (57.5%) | 20 (71.4%) | 80 (54.8%) | |
| 1B2 | 22 (12.6%) | 0 (0%) | 22 (15.1%) | |
| >1B2 | 29 (16.7%) | 1 (3.6%) | 28 (19.2%) | |
| Sedlis positive | 60 (34.5%) | 3 (10.7%) | 57 (39.0%) | **0.004** |
| Tumor size (mm, median, range) | 22.3 (no residual–79) | 12 (2.7–33) | 24 (no residual–79) | **0.005** |
| LVSI positive | 66 (37.9%) | 6 (21.4%) | 60 (41.1%) | **0.05** |
| Deep stromal invasion | 72 (41.4%) | 5 (17.9%) | 67 (45.9%) | **0.006** |
| 4-factor model positive | 77 (44.3%) | 6 (21.4%) | 71 (48.6%) | **0.008** |
| Peters criteria positive | 31 (17.8%) | 2 (7.1%) | 29 (19.9%) | 0.17 |
| LN removed (*n*, median, range) | 12 (0–32) | 11.5 (5–24) | 12 (0–32) | 0.42 |
| LN positive | 26 (14.9%) | 1 (3.6%) | 25 (17.1%) | 0.082 |
| Margin positive | 6 (3.5%) | 0 (0%) | 6 (4.1%) | 0.59 |
| Parametria positive | 7 (4.0%) | 0 (0%) | 7 (4.8%) | 0.60 |
| Adjuvant RT | 81 (46.6%) | 5 (17.9%) | 76 (52.1%) | **<0.001** |
| Adjuvant ChemoRT | 62 (35.6%) | 2 (7.1%) | 60 (41.1%) | **<0.001** |
| Interval between surgery and RT (days, median, range) | 63.5 | 61 | 64 | 0.59 |
| B | | | | |
| | **Total (174)** | **Adjuvant RT (81)** | **No RT (93)** | ***p*-Value** |
| Age (years, median, range) | 43 (22–79) | 43 (24–79) | 42 (22–77) | 0.38 |
| F/U (years, median, range) | 4.1 (0–14.8) | 5.2 (0.2–14.8) | 3.3 (0–13.8) | **<0.0001** |

**Table 1.** *Cont.*

| Histopathology | | | | 0.061 |
|---|---|---|---|---|
| Adenocarcinoma | 70 (40.2%) | 28 (34.6%) | 42 (45.2%) | |
| Squamous cell carcinoma | 94 (54.0%) | 45 (55.6%) | 49 (52.7%) | |
| Other | 10 (6.9%) | 8 (9.9%) | 2 (2.2%) | |
| Final stage | | | | **<0.0001** |
| 1A | 23 (13.2%) | 0 (0%) | 23 (24.7%) | |
| 1B1 | 100 (57.5%) | 32 (39.5%) | 68 (73.1%) | |
| 1B2 | 22 (12.6%) | 21 (25.9%) | 1 (1.1%) | |
| >1B2 | 29 (16.7%) | 28 (34.6%) | 1 (1.1%) | |
| Sedlis positive | 60 (34.5%) | 56 (69.1%) | 4 (4.3%) | **<0.0001** |
| Tumor size (mm, median, range) | 22.3 (no residual–79) | 36 (6.2–79) | 12 (no residual–55) | **<0.0001** |
| LVSI positive | 66 (37.9%) | 54 (66.7%) | 12 (12.9%) | **<0.0001** |
| Deep stromal invasion | 72 (41.4%) | 63 (77.8%) | 9 (9.7%) | **<0.0001** |
| 4-factor model positive | 77 (44.3%) | 68 (84.0%) | 9 (9.7%) | **<0.0001** |
| Peters criteria positive | 31 (17.8%) | 29 (35.8%) | 2 (2.2%) | **<0.0001** |
| LN removed (number, median, range) | 12(0–32) | 13 (3–32) | 11 (0–27) | 0.38 |
| LN positive | 26 (14.9%) | 25 (30.9%) | 1 (1.1%) | **<0.0001** |
| Margin positive | 6 (3.5%) | 4 (4.9%) | 2 (2.2%) | 0.42 |
| Parametria positive | 7 (4.8%) | 6 (7.4%) | 1 (1.1%) | 0.051 |
| **C** | | | | |
| | **Open (146)** | **Adjuvant RT (76)** | **No RT (70)** | ***p*-Value** |
| Age (years, median, range) | 43 (22–79) | 43 (24–79) | 43 (22–77) | 0.78 |
| F/U (years, median, range) | 4.3 (0.1–14.8) | 5.2 (0.2–14.8) | 3.3 (0.1–13.8) | **<0.0001** |
| Histopathology | | | | **0.187** |
| Adenocarcinoma | 52 (35.6%) | 26 (34.2%) | 26 (37.1%) | |
| Squamous cell carcinoma | 84 (57.5%) | 42 (55.3%) | 42 (60%) | |
| Other | 10 (6.8%) | 8 (10.5%) | 2 (2.9%) | |
| Final stage | | | | **<0.0001** |
| 1A | 16 (11.0%) | 0 | 16 | |
| 1B1 | 80 (54.8%) | 28 | 52 | |
| 1B2 | 22 (15.1%) | 21 | 1 | |
| >1B2 | 28 (19.2%) | 27 | 1 | |
| Sedlis positive | 57 (39.0%) | 53 (69.7%) | 4 (5.7%) | **<0.0001** |
| Tumor size (mm, median, range) | 24 (no residual–79) | 38 (6.2–79) | 12 (no residual–55) | **<0.0001** |
| LVSI positive | 60 (41.1%) | 50 (65.8%) | 10 (14.3%) | **<0.0001** |
| Deep stromal invasion | 67 (45.9%) | 60 (79.0%) | 7 (10%) | **<0.0001** |
| 4-factor model positive | 71 (48.6%) | 64 (84.2%) | 7 (10%) | **<0.0001** |
| Peters criteria positive | 29 (19.9%) | 27 (35.5%) | 2 (2.9%) | **<0.0001** |
| LN removed (number, median, range) | 12 (0–32) | 13 (3–32) | 11 (0–27) | 0.16 |
| LN positive | 25 (17.1%) | 24 (31.6%) | 1 (1.4%) | **<0.0001** |
| Margin positive | 6 (4.1%) | 4 (5.3%) | 2 (2.9%) | 0.68 |
| Parametria positive | 7 (4.8%) | 6 (7.9%) | 1 (1.4%) | 0.118 |

Of these 174 women, 28 (16%) had an MIS approach and 146 (84%) had open surgery. Patient demographics, tumor characteristics and use of adjuvant treatment was similar between the MIS and open surgery groups; there were no significant differences between groups with respect to age or follow-up. However, higher risk disease was more commonly treated with open surgery; there was significantly more Sedlis positivity, larger tumors, more LVSI positivity, deeper stromal invasion and more four-factor model positivity in the open surgery group. Interestingly, there was no difference in Peters criteria. Adjuvant RT and adjuvant chemoRT were used significantly more in patients treated with open surgery. The interval between surgery and radiotherapy was similar between patients having an MIS approach (median 61 days) and open surgery (median 64 days).

In total, 81 women received adjuvant pelvic RT (Table 1B) and those treated with RT had significantly higher risk disease; 56 (69%) met Sedlis criteria, 68 (84%) met the four-factor model and 29 (35.8%) met Peters criteria. Of those who received adjuvant pelvic RT, 76 were in the open surgery group (52% of open surgery group) and five in the MIS group (18% of MIS group). In total, 62 women had adjuvant chemoRT, with 31 (50%) meeting Peters criteria. Median dose of radiation was 45Gy in 25 fractions over 5 weeks. The median follow-up was 49 months.

As we found that women treated with open surgery had significantly higher risk disease (Table 1A), we can look further at those treated with open surgery (Table 1C). In total, 76 (52%) of the 146 women received adjuvant RT. Of these, 53 (70%) met Sedlis criteria, 62 (84%) met four-factor model and 27 (36%) met Peters criteria.

Median PFS and OS could not be estimated because majority participants did not have a recurrence or death till the last follow-up. For the whole cohort, estimated 5-year PFS probability was 83.6%, 5-year OS probability was 91.1% and the 5-year locoregional recurrence probability remained low at 9.7% (Table 2). 5-year PFS (Figure 1A) was significantly better in the open surgery compared to MIS (87.3% vs. 56.1%, $p = 0.014$). Additionally, the 5-year locoregional recurrence (Figure 2 was significantly higher in the MIS group (29.4% vs. 6.8%, $p = 0.011$) but this did not translate to an OS benefit (Figure 1B, 91.5% vs. 87.9%, $p = 0.72$). With respect to the use of adjuvant RT, there were no differences in 5-year PFS, locoregional recurrence and OS (Figure 3) with or without the use of adjuvant RT ($p = 0.54$, 0.59 and 0.25, respectively).

**Table 2.** Five-year OS, PFS and locoregional recurrence as a function of surgical approach and the use of adjuvant RT.

| | 5-Year OS (%) | *p*-Value | 5-Year PFS (%) | *p*-Value | 5-Year Locoregional Recurrence (%) | *p*-Value |
|---|---|---|---|---|---|---|
| Total (174) | 91.1% | | 83.6% | | 9.7% | |
| MIS (28) | 87.9% | 0.72 | 56.1% | 0.014 | 29.4% | 0.011 |
| Open (146) | 91.5% | | 87.3% | | 6.8% | |
| Adjuvant RT (81) | 88.5% | 0.25 | 82.4% | 0.54 | 8.2% | 0.59 |
| No RT (93) | 94.7% | | 83.8% | | 12.4% | |

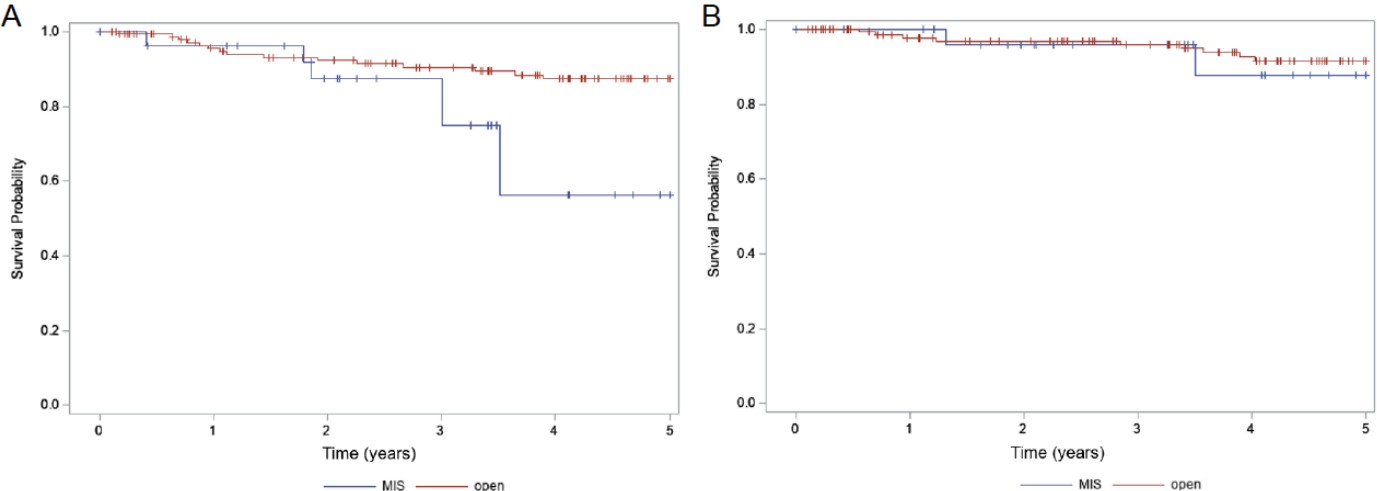

**Figure 1.** Kaplan–Meier survival curves as a function of surgery with (**A**) 5-year PFS 56.1% in MIS vs. 87.3% in open (*p* = 0.014); (**B**) 5-year OS 87.9% in MIS and 91.5% in open (*p* = 0.72).

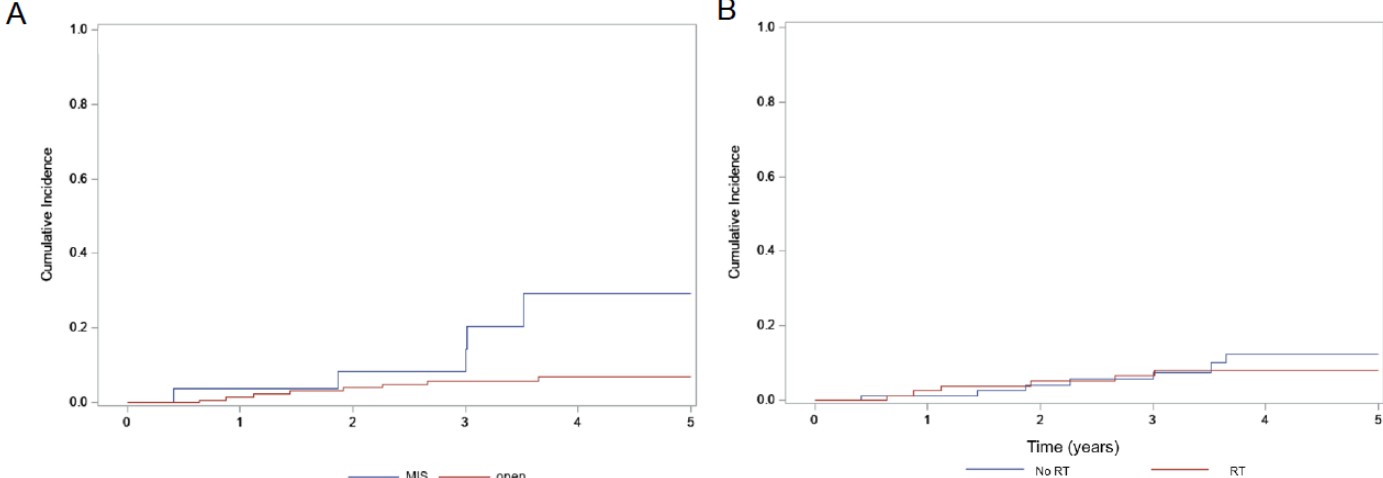

**Figure 2.** Five-year locoregional recurrence rates as a function of intervention with (**A**) 5-year locoregional recurrence significantly higher with MIS at 29.4% vs. open surgery at 6.8% (*p* = 0.011); (**B**) 5-year locoregional recurrence rate of 8.2% with adjuvant RT vs. 12.4% without RT, but this is not statistically significant (*p* = 0.59).

There were 14 (8%) perioperative complications, including readmission to hospital (9; 5.2%) and intra-operative complications (3, 1.7%). Complications were exclusively bleeding requiring transfusions, wound dehiscence and post-operative infections. All nine readmissions were due to wound dehiscence or infections. Two patients (1.1%) required repeat surgery within 30 days and one patient (0.6%) was admitted to ICU or a short period. There was no statistically significant difference in perioperative complications between MIS and open surgery. There were nine (5.2%) postoperative complications, with no statistically significant difference between the MIS and open surgery group (*p*-value = 0.16).

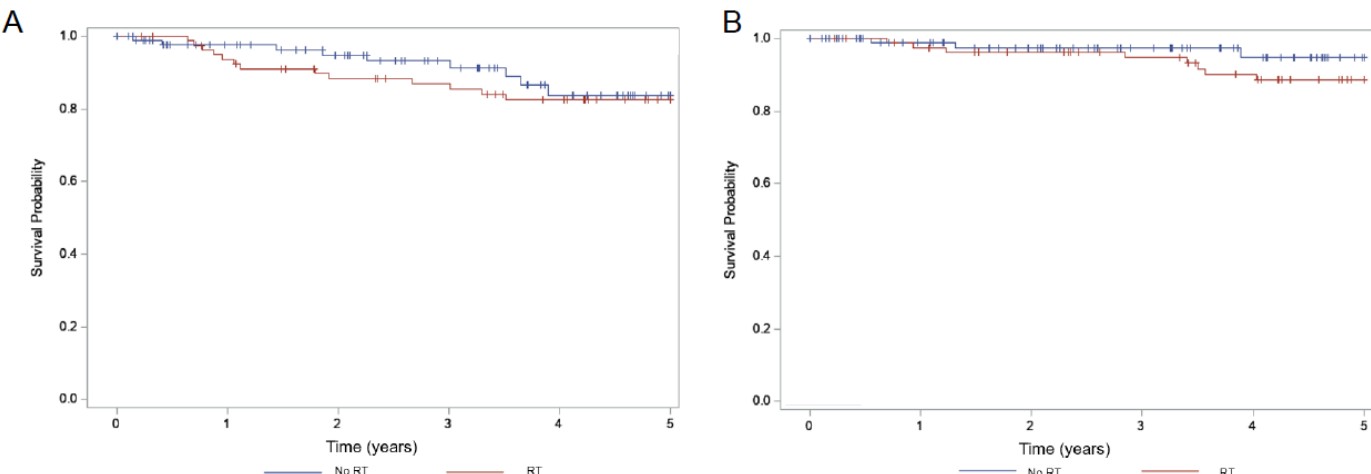

**Figure 3.** Kaplan–Meier survival curves as a function of use of adjuvant RT with (**A**) 5-year PFS 82.4% with RT and 83.8% without RT (*p* = 0.54); (**B**) 5-year OS 88.5% with RT and 94.7% without RT (*p* = 0.25).

## 4. Discussion

We have demonstrated excellent long-term outcomes for women treated for early-stage cervix cancer, with greater than 90% locoregional control with selective use of RT (with or without chemotherapy), even among patients with adverse pathologic features.

While the 5-year locoregional recurrence rate was only 9.7% in the 174 women treated with surgery with or without adjuvant therapy, there was a significant difference in this locoregional recurrence rate with surgical approach. There was a significantly higher recurrence rate with an MIS approach (6.8% with open vs. 29.4% with MIS *p* = 0.011). These findings are consistent with a recently published prospective, randomized controlled trial of 631 women with early-stage cervix cancer that randomized to MIS or open surgery [5]. At 4.5 years, DFS was 86.0% in the MIS group and 96.5% in the open surgery group (*p* = 0.87 for non-inferiority). MIS was also associated with lower 3-year OS (93.8% vs. 99.0%, HR 6.00) and a higher 3-year rate of death from cervix cancer (4.4% vs. 0.6%, HR 6.56). Similarly, a cohort study using the Surveillance, Epidemiology and End Results program database including 2461 women with early-stage cancer found that after the introduction of MIS, there was a decrease in 4-year survival of 0.8% per year (*p* = 0.01) [6]. Moreover, they found that the 4-year mortality rate for MIS was 9.1% compared to 5.3% with open surgery (*p* = 0.002). Various theories for the survival discrepancies between MIS and open approaches have been suggested including the uterine manipulator used in MIS having an increased propensity for tumor spillage or the insufflation gas used in MIS causing tumor seeding [11,12]. In fact, a multicenter retrospective observational cohort study investigated outcomes in patients with stage IB1 cervix cancer treated with MIS and open surgery [13]. They found that the risk of recurrence in those treated with MIS was twice as high as in open surgery and the risk of death was 2.4 times higher in the MIS group. When patients had MIS with uterine manipulator they had a 2.76 times higher hazard of relapse, but when patients had MIS without the uterine manipulator or with protective vaginal closure, their rates of relapse were similar to in the open surgery group. This lends support to the idea that the manipulator may be a source of recurrence.

It should be noted that, while we found a significantly higher locoregional recurrence and worse PFS with MIS compared to open surgery, this did not translate to a survival benefit like in the studies described above. This is likely secondary to the small sample size, with only 28 women having undergone an MIS approach, compared to 146 women having undergone an open surgery. Additionally, some recurrences were salvaged with additional therapy. Unfortunately, a significant proportion of the women in our study who underwent MIS experienced a recurrence, despite having relatively lower risk disease.

Only 10.7% of those treated with MIS met Sedlis criteria, 21.4% met 4-factor model criteria and 7.1% met Peters criteria. Of those who underwent MIS, 17.9% received adjuvant RT and all those meeting Peters criteria received chemoRT. Too few patients received RT to draw conclusions about outcomes with and without adjuvant RT in the MIS group.

The OS and PFS of women in our study (5-year OS 91.1% and 5-year PFS 83.6%) was shorter than the aforementioned studies, but there are several factors that may explain this. First, we included slightly higher stage disease with some patients having stage 1B2 or even higher stage cervix cancer, found at the time of pathology (28.7%), but these studies were limited to 1A1, 1A2 or 1B1 cervical cancer. Additionally, the 5-year timepoint used in our study was later than in these studies (3-year and 4-year, respectively) with the former having median follow-up of 2.5 years and median follow-up in our study was nearing 5 years. Additionally, Ramirez et al. had no significant differences between open surgery and MIS groups with respect to high-risk features, whereas the women in our study who underwent open surgery had significantly higher risk disease ($p < 0.001$), which may have played a role. Moreover, the use of RT and/or chemotherapy was well-balanced in their study and much lower than in ours, 27.6% and 28.8% in open and MIS [5], respectively, compared to 52.1% and 17.9% in ours.

The higher use of adjuvant treatment in our study was not overuse, but instead, secondary to higher risk features. Of those who received adjuvant RT, 69.7% met Sedlis criteria [7], 84.2% met 4-factor model, where adenocarcinoma is also incorporated as a high-risk feature [10] and 35.5% met Peters criteria [8]. This demonstrates that use of radiotherapy was selective and applied to higher risk disease. Importantly, there were no significant differences in 5-year locoregional recurrence, 5-year PFS or 5-year OS with the use of adjuvant RT. This is impressive given that these patients tended to have higher risk disease compared to those not receiving radiation. This provides further support for continued use of selection criteria like Sedlis, four-factor model and Peters criteria to continue to improve outcomes in early-stage cervix cancer. Melamed et al. did find, similar to our study, that women who underwent MIS tended to have smaller, lower grade tumors than with the open approach [6].

Our results are also consistent with other studies with long-term follow-up. Albeit, with patients with higher risk disease, Landoni et al. randomized 346 women with stage IB–IIA cervical carcinoma to radical surgery or RT. Twenty-year OS was 72% and 77% for surgery and RT, respectively ($p = 0.28$) [14]. They found that 28.2% of women treated with surgery experienced recurrences and 27% of those treated with RT experienced recurrences, with median time to relapse 13.5 months and 11.5 months ($p = 0.01$), respectively. Looking specifically at patients with stage IB cervical cancer and higher risk features, a prospective phase III randomized trial randomized women to adjuvant RT if they had stage IB cervical cancer and two or more of one third stromal invasion, LVI or tumor size $\geq 4$ cm [9]. After a median follow-up of 10 years, 17.5% of patients experienced recurrences, consistent with our data. The slightly higher numbers may be secondary to the fact that chemotherapy was not used in their study. This compares to Peters et al. where women were randomized to adjuvant RT or chemoRT and 4-year PFS was 80% in women treated with chemoRT [8]. These women did have higher risk disease, being required to meet Peters criteria; one of positive lymph node, positive margin or positive parametria. Our data are also in line with a number of retrospective reviews from single and multiple institutions, as well as a number of cohort studies with similar early-stage cervix cancer patient population do show similar DFS and OS outcomes [15–19].

A large reason for the movement towards an MIS approach was the promise of lower rates of complication. Retrospective reviews [20–22] have shown fewer intraoperative and post-operative complications, as well as shorter hospital stays, but we found no significant difference between the MIS and open approaches. It is possible that this is secondary to the overall low rate of complications and the relatively few women who underwent an MIS approach.

There are some limitations of our study. First, it is a retrospective study and limited to a single institution. There is excellent long-term follow-up; however, there is shorter follow-up in patients with lower risk disease, and this population overlaps with those who did not receive adjuvant RT. Moreover, while the National Comprehensive Cancer Network guidelines incorporated MIS as an option for hysterectomy [23], the use at our institution was not significant and so the MIS arm is much smaller than the open surgery arm of our study.

## 5. Conclusions

We have demonstrated that excellent long-term outcomes are achievable in the treatment of early-stage cervix cancer. The patients who received adjuvant RT often tended to have adverse pathologic features, yet still did quite well with similar PFS, locoregional recurrence and OS to those where adjuvant RT was omitted. Some have suggested de-escalation of treatment for patients who meet Sedlis criteria, with improved surgical techniques compared to when Sedlis was originally published, and advocate for a personalized approach [24] and the excellent outcomes herein may support this. This is being investigated in the CERVANTES study currently [25] and may allow for de-escalation in these patients.

Our data also demonstrate that MIS surgery should be avoided as it is associated with worse PFS and higher rates of locoregional recurrence. However, some hypothesize that the outcomes with an MIS approach are caused by laparoscopic surgery and outcomes may be excellent for robotic-assisted surgery. Trials are underway with a robotic approach alone incorporating vaginal protective measures [26,27] and we await these results.

**Author Contributions:** Conceptualization, R.S., T.L. and L.B.; methodology, L.B. and W.A.; formal analysis, T.Z.; data curation, L.B. and W.A.; writing—original draft preparation, L.B.; writing—review and editing, W.A., T.Z., K.L., M.F.-K.-F., W.F., T.L. and R.S.; supervision, R.S. All authors have read and agreed to the published version of the manuscript.

**Funding:** This research received no external funding.

**Institutional Review Board Statement:** The study was conducted in accordance with the Declaration of Helsinki, and approved by the Institutional Review Board of Ottawa Health Sciences Network Research Ethics Board (protocol code 20190390-01H and approved July 2019).

**Informed Consent Statement:** Patient consent was waived as this was a retrospective review.

**Data Availability Statement:** Data available upon request, due to restrictions.

**Conflicts of Interest:** The authors declare no conflict of interest.

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
