# Peer review of "Evaluation of Surgical Approaches and Use of Adjuvant Radiotherapy with Respect to Oncologic Outcomes in the Management of Clinically Early-Stage Cervical Carcinoma"

_curroncol, doi:10.3390/curroncol29120748_

Round 1

Reviewer 1 Report

The paper entitled “Evaluation of Surgical Approaches and Use of Adjuvant Radiotherapy with Respect to Oncologic Outcomes in the Management of Early-Stage Cervical Carcinoma” by Laura Burgess et al. reports the retrospective analysis of disease outcome in a series of 174 women with early-stage invasive cervical carcinoma. It shows that a 90% rate of overall survival could be obtained in these patients with a combination of surgery and adjuvant therapy. The paper shows in addition that a higher rate of relapse was observed in patients that benefited of minimally invasive surgery than in patients with open surgery. The outcome was similar in patients with or without adjuvant radiotherapy.

The study is well conducted, the methodology sound and the results significant. It shows that disease outcome remained favorable when adjuvant radiotherapy was applied in patients presenting adverse pathological features.

My main remark concerns the statement in the conclusion that “The use of adjuvant RT seems to make up for adverse pathologic features, as these women have similar PFS, locoregional recurrence and OS to those where adjuvant RT was omitted, despite adverse pathologic features. It was not clear for me that adjuvant RT had been omitted in a subgroup of patients with adverse criteria. For instance, it is stated that the rate of patients with > IB2 stage was 34% VS 1% in patients with or without radiotherapy. I understand from the study that radiotherapy has the potential to correct the negative effect of adverse pathological feature. Is this correct? The authors seem to suggest in the conclusion that the indication of adjuvant radiotherapy could be reduced. If this is the case, could the author better detail in the results the arguments for this statement. For instance, what was the outcome in the group of patients with adverse pathological features that did not received radiotherapy, if there was some. Or I have not well understood the sentence in the conclusion that should be more clearly explained.

Minor remark : the last sentence has to be corrected.

Author Response

The paper entitled “Evaluation of Surgical Approaches and Use of Adjuvant Radiotherapy with Respect to Oncologic Outcomes in the Management of Early-Stage Cervical Carcinoma” by Laura Burgess et al. reports the retrospective analysis of disease outcome in a series of 174 women with early-stage invasive cervical carcinoma. It shows that a 90% rate of overall survival could be obtained in these patients with a combination of surgery and adjuvant therapy. The paper shows in addition that a higher rate of relapse was observed in patients that benefited of minimally invasive surgery than in patients with open surgery. The outcome was similar in patients with or without adjuvant radiotherapy.

The study is well conducted, the methodology sound and the results significant. It shows that disease outcome remained favorable when adjuvant radiotherapy was applied in patients presenting adverse pathological features.

My main remark concerns the statement in the conclusion that “The use of adjuvant RT seems to make up for adverse pathologic features, as these women have similar PFS, locoregional recurrence and OS to those where adjuvant RT was omitted, despite adverse pathologic features. It was not clear for me that adjuvant RT had been omitted in a subgroup of patients with adverse criteria. For instance, it is stated that the rate of patients with > IB2 stage was 34% VS 1% in patients with or without radiotherapy. I understand from the study that radiotherapy has the potential to correct the negative effect of adverse pathological feature. Is this correct? The authors seem to suggest in the conclusion that the indication of adjuvant radiotherapy could be reduced. If this is the case, could the author better detail in the results the arguments for this statement. For instance, what was the outcome in the group of patients with adverse pathological features that did not received radiotherapy, if there was some. Or I have not well understood the sentence in the conclusion that should be more clearly explained.

Response: Our apologies for the lack of clarity. We have reworded this instead to say that patients who received adjuvant RT often had adverse pathologic features, but despite these adverse features, they had similar PFS, OS and locoregional control to the patients who did not receive adjuvant RT (and did not receive adjuvant RT because they tended not to have such adverse pathologic features). There was no group with adverse pathologic features where radiotherapy was withheld. Thank you for the remark, as we feel that the concluding statement is now clearer.

Minor remark : the last sentence has to be corrected.

Response Thank you for noting the duplicate. We have now corrected this and appreciate your notation.

Reviewer 2 Report

The subject of the reviewed paper was the evaluation of surgical methods and the use of adjuvant radiotherapy in relation to oncological outcomes in the treatment of early stage cervical cancer. The authors put a lot of work into presenting reliable research and observations. The work is written clearly and understandably. I have no comments regarding the quality of work. I consider it fit for publication in Current Oncology.

Author Response

The subject of the reviewed paper was the evaluation of surgical methods and the use of adjuvant radiotherapy in relation to oncological outcomes in the treatment of early stage cervical cancer. The authors put a lot of work into presenting reliable research and observations. The work is written clearly and understandably. I have no comments regarding the quality of work. I consider it fit for publication in Current Oncology.

Response: Thank you. We appreciate your remarks.

Reviewer 3 Report

The authors of the study performed a retrospective analysis of data from 174 patients with cervical cancer who underwent radical surgery (open or MIS) with or without adjuvant radio(chemo)therapy. The aim of the study was to assess the treatment results (OS, PFS, 5-year locoregional recurrence) and the impact of the surgical technique on the decision on adjuvant treatment and survival. In conclusion, the authors stated that PFS was significantly lower and 5-year locoregional recurrence was significantly higher  in patients who had MIS vs. open surgery and which suggests that MIS should be avoided.

Below are some comments that may contribute to greater transparency of the work.

1. Regardless of the fact that the authors used the 2009 FIGO classifications, the accuracy of the title of the work raises doubts. Considering that 25 patients did not meet the classical criteria for early cervical cancer due to metastases to the parametria and/or lymph nodes it is worth changing it to e.g.: „Evaluation of surgical approaches and use of adjuvant radiotherapy with respect to oncologic outcomes in the management of patients with intermediate and high risk cervical carcinoma”.

2. Although open surgery is recommended as the standard method (this aspect was discussed by the authors in the discussion), for the purposes of the article Part 2.1 (Design) should include at least a brief mention of imaging tests that are the basis for proper qualification for the use of a specific surgical technique.

3. Regarding table 1A,B,C:

The group meeting the 4-factor model positive criteria (77 pts) should be characterized analogously to Sedlis positive.

A group of 29 patients with stage >1B2 needs to be supplemented - what stages are we talking about?

The total group of patients is 174, and the sum of Sedlis positive (60 pts) + 4factor model (77 pts) + Peters criteria positive (31 pts) = 168 ?

Table 1A shows that a total of 143 (81+62) pts received RT/ChemoRT which means that 31pts (174 -143) did not receive adjuvant treatment, and in table 1B we can see  that 93 patients did not receive radiotherapy ?

Assuming that 143 patients received adjuvant treatment and only 31 did not, the lack of statistical “evidence” for the beneficial effect of adjuvant treatment may be due to the small size of the latter group (figure2).

Author Response

The authors of the study performed a retrospective analysis of data from 174 patients with cervical cancer who underwent radical surgery (open or MIS) with or without adjuvant radio(chemo)therapy. The aim of the study was to assess the treatment results (OS, PFS, 5-year locoregional recurrence) and the impact of the surgical technique on the decision on adjuvant treatment and survival. In conclusion, the authors stated that PFS was significantly lower and 5-year locoregional recurrence was significantly higher  in patients who had MIS vs. open surgery and which suggests that MIS should be avoided.

Below are some comments that may contribute to greater transparency of the work.

1. Regardless of the fact that the authors used the 2009 FIGO classifications, the accuracy of the title of the work raises doubts. Considering that 25 patients did not meet the classical criteria for early cervical cancer due to metastases to the parametria and/or lymph nodes it is worth changing it to e.g.: „Evaluation of surgical approaches and use of adjuvant radiotherapy with respect to oncologic outcomes in the management of patients with intermediate and high risk cervical carcinoma”.

Response: Thank you for the suggestion. We would not consider these patients high-risk per se, and all were felt to be clinically early-stage. We do agree that given some parametrial and lymph node involvement, they did not all end up as pathologically early-stage and have renamed the titled using “clinically early-stage” and appreciate the suggestion.

2. Although open surgery is recommended as the standard method (this aspect was discussed by the authors in the discussion), for the purposes of the article Part 2.1 (Design) should include at least a brief mention of imaging tests that are the basis for proper qualification for the use of a specific surgical technique.

Response: We did not mention the pre-operative staging and we appreciate your suggestion that it be included. As such, we have now added details of the fact that most patients did not routinely undergo pre-operative CT AP or MRI pelvis, as per local institutional practice.

3. Regarding table 1A,B,C:

The group meeting the 4-factor model positive criteria (77 pts) should be characterized analogously to Sedlis positive.

Response: After discussions with our statistician, we do not believe that this further analysis, though interesting, would add much to our results, analyses or conclusions (in a meaningful way).  Therefore, we respect the comments and suggestions, but we do not believe it is necessary at this time.

A group of 29 patients with stage >1B2 needs to be supplemented - what stages are we talking about?

Response: We agree that additional detail should be provided and as such we have included this. Four of these patients had pT2aN0 and the others were node positive. We have now included this in the manuscript and appreciate the recommendation.

The total group of patients is 174, and the sum of Sedlis positive (60 pts) + 4factor model (77 pts) + Peters criteria positive (31 pts) = 168 ?

Response: To clarify, each row in the table is not exclusive from one another. The same patient may have been Sedlis positive, 4-factor model positive and Peters positive, but of the entire group of 174 patients, 60 were Sedlis positive, 77 were 4-factor model positive and 31 were Peters positive. We apologize for the lack of clarity and have now added further description in the table heading to clarify.

Table 1A shows that a total of 143 (81+62) pts received RT/ChemoRT which means that 31pts (174 -143) did not receive adjuvant treatment, and in table 1B we can see  that 93 patients did not receive radiotherapy ?

Response: Again, we apologize for the lack of clarity and have added qualifying statements on to the table headings. Similarly, to the previous comment, the rows are not exclusive from one another. 81 patients received adjuvant RT and of these, 62 received chemoRT. Table 1B does in fact represent the number who received RT (81) and those who did not (174 – 81 – 93).

Assuming that 143 patients received adjuvant treatment and only 31 did not, the lack of statistical “evidence” for the beneficial effect of adjuvant treatment may be due to the small size of the latter group (figure2).

Response: We apologize again for the lack of clarity and have added the qualifying statements. In light of the fact that 93 patients did not receive RT and 81 did, we feel that the sample size is not too small and, as such, the lack of statistically significant OS difference between use of adjuvant RT and no use of adjuvant RT is in fact real. 

Reviewer 4 Report

Real life experiences are always important in confirming or disproving the results of clinical trials. It is an important result to demonstrate that minimally invasive surgery is associated with higher odds of recurrence in the context of cervical cancer

Just few suggestions and comments.

line 46

the symbol “:” should be used

lines 42 52

I think the prognostic models need to be briefly explained more precisely

lines 55-57

please clarify the second part of “ aim of the study” period

lines 62 65

please simplify the period avoiding repetitions

lines 65-66

I think that “was” is better than was left because it’s a retrospective study

Lines 151-155

can you describe more precisely or resume in a table what complications there were ? what grade ?

Survival curves

survival curves cross each other; what statistical test was used to compare the curves?

Discussion

What factor could explain the lack of difference in overall survival between MIS and open surgery groups? Do you think it is exclusively due to the small sample of MIS women?

Author Response

Real life experiences are always important in confirming or disproving the results of clinical trials. It is an important result to demonstrate that minimally invasive surgery is associated with higher odds of recurrence in the context of cervical cancer

Just few suggestions and comments.

line 46

the symbol “:” should be used

Response: Thank you for noting this. It has now been corrected

lines 42 52

I think the prognostic models need to be briefly explained more precisely

Response: We agree that this warrants additional description and appreciation he recommendation. We have now expanded the discussion of the prognostic models and the potential benefit of use of RT and chemoRT in each case.  

lines 55-57

please clarify the second part of “ aim of the study” period

Response: We agree that this could have been clearer and appreciate the suggestion. We have clarified the secondary aims.

lines 62 65

please simplify the period avoiding repetitions

Response: While originally the study period was described in a more complicated manner, upon your advice, we have reworded the study period and believe it is now clearer and without repetition.

lines 65-66

I think that “was” is better than was left because it’s a retrospective study

Response: Thank you, this was a retrospective review so we could not leave decisions to the surgeon based on the study.

Lines 151-155

can you describe more precisely or resume in a table what complications there were ? what grade ?

Response: We agree that there should have been greater detail in the discussion around the complications. We have now added in further details of the complications; bleeding requiring transfusions, wound dehiscence and infection. All readmissions were due to wound dehiscence or infection. We have also specified repeat surgeries required and ICU admissions required.  

Survival curves

survival curves cross each other; what statistical test was used to compare the curves?

Response: We have elaborated this in 2.2 Analysis at your suggestion. We appreciate the suggestion, as the statistical methods were not clear in our original manuscript. We clarified that we looked at 5-year progression-free survival and overall survival and that log-rank test for statistical significance was used.

Discussion

What factor could explain the lack of difference in overall survival between MIS and open surgery groups? Do you think it is exclusively due to the small sample of MIS women?

Response: We believe that the lack of overall survival difference between MIS and open is likely secondary to the sample size. This would explain the discordance in OS results when compared to the studies by Ramirez and Melamed. Other possibilities include salvage of relapses with other therapies. We have added this to the discussion and appreciate the recommendation.